# Journal editors' perspectives on the roles and tasks of peer reviewers in biomedical journals: a qualitative study

Ketevan Glonti ,[1,2] Isabelle Boutron ,[2] David Moher ,[3] Darko Hren [1]

¹Department of Psychology, School of Humanities and Social Sciences, University of Split, Split, Croatia
²CRESS, INSERM, INRA, Université de Paris, Paris, France
³Ottawa Methods Centre, Ottawa Hospital Research Institute, Ottawa, Ontario, Canada

**Correspondence to**
Ketevan Glonti; kglonti@unist.hr

## ABSTRACT

**Objective** Peer reviewers of biomedical journals are expected to perform a large number of roles and tasks, some of which are seemingly contradictory or demonstrate incongruities between the respective positions of peer reviewers and journal editors. Our aim was to explore the perspectives, expectations and understanding of the roles and tasks of peer reviewers of journal editors from general and specialty biomedical journals.

**Design** Qualitative study.

**Setting** Worldwide.

**Participants** 56 journal editors from biomedical journals, most of whom were editors-in-chief (n=39), male (n=40) and worked part-time (n=50) at journals from 22 different publishers.

**Methods** Semistructured interviews with journal editors were conducted. Recruitment was based on purposive maximum variation sampling. Data were analysed thematically following the methodology by Braun and Clarke.

**Results** Journal editors' understanding of the roles and partly of tasks of peer reviewers are profoundly shaped by each journal's unique context and characteristics, including financial and human resources and journal reputation or prestige. There was a broad agreement among journal editors on expected technical tasks of peer reviewers related to scientific aspects, but there were different expectations in the level of depth. We also found that most journal editors support the perspective that authorship experience is key to high-quality reviews, while formal training in peer reviewing is not.

**Conclusion** These journal editors' accounts reveal issues of a social nature within the peer-review process related to missed opportunities for journal editors to engage with peer reviewers to clarify the expected roles and tasks. Further research is needed on actual performance of peer reviewers looking into the content of peer-reviewer reports to inform meaningful training interventions, journal policies and guidelines.

## INTRODUCTION

Peer reviewers of biomedical journals are key stakeholders in the editorial ecosystem, helping authors to improve manuscripts and providing advice to scientific editors on their decision regarding the acceptability of publishing papers. Despite their importance for scientific publishing, fundamental principles such as the roles, tasks and core

### Strengths and limitations of this study

► This study is one of few qualitative studies that explore biomedical journal editors' views regarding the roles and tasks of peer reviewers.
► The participants were diverse in terms of characteristics related to the journals.
► The majority of the participants were editors-in-chief, which may limit the generalisability of the results.

competencies of peer reviewers—including a minimum standard of knowledge, skills and characteristics that are needed to effectively deliver high-quality reviewer reports—are neither well defined, agreed on nor formally established.[1] While core competencies have been to some degree established for journal editors,[2] thus far, this is not the case for peer reviewers. A recent scoping review (2019) showed a large number of roles and tasks that peer reviewers of biomedical journals are expected to carry out, some of which seem to contradict each other or display incongruities between the position of the peer reviewer and the position of the journal editor.[3] These findings were reflected in a study that aimed to identify the tasks that journal editors expect from peer reviewers who evaluate a manuscript reporting a randomised controlled trial, where a substantial disconnect between the expectations of journal editors and peer reviewers was found.[4] A mutual understanding of expectations and responsibilities is one of the key factors that determine the quality of reviewer reports and satisfaction of the actors with the review process. However, biomedical journals differ in their guidance provided to peer reviewers, in their publishing capacity and resources available, as well as the reviewer pool.[5] Therefore, it is likely that journal editors might have diverging opinions about the roles and tasks peer reviewers are supposed to perform, something that has not been previously explored in depth.

Given that peer review is a social process that goes beyond the quality control of manuscripts,[6] qualitative methods may lead to a deeper examination of the complexities of these processes compared with quantitative approaches and may provide important context to improve the understanding of different editorial realities and practices.

Our aim was to examine the experience of general and specialty biomedical journal editors and to characterise their perspectives, expectations and understanding of the roles and tasks of peer reviewers.

## METHODS
### Study design
We conducted semistructured interviews with biomedical journal editors from general and specialty journals. The design of the study and reporting of study results were informed by relevant guidance for reporting qualitative research.[7] Key methodological components are presented further; a detailed description of the study methodology is available elsewhere.[8]

### Patient and public involvement
Patients and the public were not involved in the design, conduct, reporting or dissemination of our research, sampling and recruitment.

### Sampling and recruitment
We used purposive maximum variation sampling[9] to obtain as much diversity in the demographic and journal characteristics of study participants as possible. Interviewees were recruited from multiple sources, including the lead author's professional network within the Methods in Research on Research project[10]; from two publishers, namely, BioMed Central and British Medical Journal publishing group; and attendees of the Eighth International Congress on Peer Review and Scientific Publication.[11] A total of 543 prospective interviewees were approached via email, and 69 journal editors responded positively to the request. In addition, interviewees were asked to recommend other editors who would potentially be interested in contributing to this study.

Since sample size is irreversibly linked to saturation, which in turn can only be operationalised during data collection,[12] our approach to data collection and analysis was iterative. Thus, recruitment continued until saturation—conceptualised as the point at which no new codes and themes were identified from the data—was achieved. After 56 interviews, saturation was obtained and no further journal editors were contacted and interviewed.

### Data collection
All interviews were conducted between October 2017 and February 2018 by the lead author (KG). Interviews were conducted either face-to-face or by telephone to accommodate for the geographical diversity and availability of study participants. They lasted 25–60 minutes.

A topic guide (online supplementary additional file 1) was used during the semistructured interviews. The guide was initially informed by the outcomes of the scoping review[1] and was piloted and further refined over the course of the study, particularly after the first four interviews.

Prospective interviewees were provided with a study consent form and a study information sheet that consisted of information about the researchers and study information (aim, interview procedures, ethics, confidentiality, funding and contact details). Interviewees were asked to sign a written consent form prior to being interviewed. Before starting the interview, study objectives were reiterated and additional information was provided where necessary.

KG was a PhD student at the time of the interviews. She has previously experienced the peer-review process in biomedical journals as an author and peer reviewer and had undergone training in conducting qualitative interviews prior to data collection. She was supervised by DH, who has extensive experience of the peer-review process in biomedical journals as an author, peer reviewer and journal editor.

### Analysis
Interviews were transcribed verbatim and fieldnotes were written up after every interview.

All documents were then imported into NVivo V.12 and were subjected to thematic analysis, as described by Braun and Clarke,[13] and outlined in the protocol.[8] In summary, a preliminary codebook was generated by two researchers (KG and DH) independently from a subset of six interviews[14] using both deductive codes from topics in the interview guide and inductive content-driven codes. The remaining 50 interviews were coded by the lead researcher (KG), supervised by DH through regular meetings. In line with the iterative process of data collection and analysis, interviews were analysed in the order in which they were conducted. To assess saturation, the lead researcher documented the process of code development, updating the codebook after analysing each transcript. Saturation was achieved after 56 interviews.

To establish trustworthiness in this research, the step-by-step approach proposed by Nowell *et al*, which provides a detailed description of how to conduct a trustworthy thematic analysis, was followed.[15] This approach used criteria for trustworthiness in qualitative research proposed by Lincoln and Guba [16] show how these can be achieved throughout the six phases of thematic analysis. The methodological techniques that we undertook to ensure a trustworthy analysis throughout our study are presented in online supplementary additional file 2.

## RESULTS
A total of 56 biomedical journal editors were interviewed (table 1). Of these, the majority were male editors-in-chief who were based in 21 different countries. Most

**Table 1** Sample characteristics

| Demographic characteristics | |
|---|---|
| Sex | Female (n=16), male (n=40) |
| Position | Junior editor (n=1), senior/associate editor (n=11), coeditor-in-chief (n=4), editor-in-chief (n=39), editorial director (n=1) |
| Commitment | Part-time (n=50), full-time (n=6) |
| Geographical location | Asia (n=2), Africa (n=1), North America (n=19), South America (n=3), Europe (n=28), Oceania (n=3) |
| **Journal characteristics** | |
| Journal specialty | General medicine and mega journals* (n=13), specialty (n=43) |
| Indexing status† COPE membership | Yes (n=53), no (n=3) Member (n=27), not a member (n=29) |
| Peer-review model | Single-blind (n=38), double-blind (n=7), triple-blind (n=1), open peer review (n=9), postpublication (n=1) |
| Open access, subscription, mixed | Open access (n=35), subscription (n=4), mixed (n=17) |
| Publishers | Academic (n=9), commercial (n=34), mixed model‡ (n=13) |

*A peer-reviewed academic open-access journal designed to be much larger than a traditional journal by exercising low selectivity among accepted articles.
†Refers to indexing status on MEDLINE, Scopus and Web of Science.
‡Refers to journals that are either co-owned by medical societies and commercial publishers, or owned entirely by medical societies but operated through a commercial publisher.
COPE, Committee on Publication Ethics.

journal editors worked part-time at their respective journals, which were mainly specialty journals. Most journals employed a single-blind review process. Most interviewees were editors of journals that were published through commercial publishers.

An overview of the different domains within our two themes (roles of peer reviewers and tasks of peer reviewers) are presented in online supplementary additional file 3.

### Roles of peer reviewers

Journal editors outlined a variety of roles, which coalesced around four domains. Peer reviewers should be (1) proficient experts in their field qualified to peer review, (2) dutiful towards the scientific community versus volunteers who deserve recognition, (3) professionals and (4) advisors to the editor.

### Peer reviewers should be 'proficient experts in their field qualified to peer review'

There was agreement among journal editors that peer reviewers are experts in their field when they (1) have expertise and demonstrate high-level knowledge in their subject area, (2) are up to date with existing evidence and practice guidelines and (3) have experience of publishing their own research. However, there was substantial disagreement on how these criteria are defined and understood and how 'expertise' is operationalised.

One common narrative was that qualified peer reviewers are 'experienced authors' who have a strong reputation and publication record in 'high-impact journals'. Concurrently, a number of journal editors linked the quality of the peer-review report with the reviewers' writing and analytical skills, which they believed are gained through extensive authorship in their field. In their view, authorship hones both writing and reviewing ability, since authors are theoretically able to learn from review reports on their own submitted manuscripts:

> You learn by doing and if you have published let's say 200 articles then normally you are also a good reviewer… and if you are a bad author of manuscripts then you are a bad reviewer. And your opinion leaders are the sought after reviewers because they know the field and can write well and can also analyse a manuscript from another author quite well. (Editor-in-chief, specialty journal).

Interviewees also indicated that they had a preference for seasoned authors and opinion leaders in the field over junior researchers. Here, their main concern was about fulfilling authors' expectations of an objective peer review by recruiting an expert to review their manuscript:

> Well first of all I think our reviewers … are seasoned, they have to be experts, I mean otherwise why are they reviewing? That is not fair to the author. (Coeditor-in-chief, specialty journal)

However, several journal editors commented that the actual level of expertise needed to deliver a high-quality review report does not necessarily depend on publication record and seniority level. Some journal editors considered reviewing to require a different type of skill set that is not necessarily developed through writing or present

by default. Other key factors drive review quality, such as 'dedication of sufficient time' and 'hands-on experience with the methods used'. This is often the case with junior researchers, who go through an active learning experience of applying methods for their own research and receiving feedback on their work. Less experienced researchers' greater motivation to peer review was also mentioned as a major driver of high-quality reviewer reports. For these reviewers, receiving the invitation to review is in itself a confirmation of growing personal reputation and recognition by the journal and by the broader scientific community. At the same time, their supposed lack of self-confidence due to their current low career status/standing within the scientific field could also drive the delivery of high-quality reviewer reports in a desire to establish and maintain their status within the scientific community:

> I will say that junior faculty and post doctorate fellows often write the best reviews because they tend to be insecure and tend to over-compensate and to be very careful in doing a good job. (Editor-in-chief, specialty journal).

In the same vein, a number of journal editors from non-high-ranking journals commented that senior reviewers' increasing scientific status and 'self-regard' might lead to declining review report quality, most commonly demonstrated by the 'lack of detailed comments' or 'two-line' review reports that did not aim to help 'to improve a manuscript', but only to judge publication potential. That being said, 'experienced' peer reviewers were still highly sought after by all interviewees. Since they typically receive a high volume of reviewer requests, journal editors suspect that they prioritise their reviewing time in favour of highly ranked journals, a behaviour that multiple journal editors reported practising themselves when asked to perform a peer review. Although the least experienced reviewers are generally more available, most editors feel that they lack the degree of experience required to conduct a good peer review and 'focus excessively on technical details', instead of the 'bigger picture' that more experienced reviewers are able to provide.

Regardless of preference for the type of peer reviewer, the vast majority of interviewees—except for those journal editors working for high-ranking journals—acknowledged that it is hard to solicit peer reviewers in general, particularly experienced ones:

> And one of the things that we face is that we have on one side younger investigators, willing to do the job. Sometimes they lack you know, the view and then you will have the very established scientist who in most cases do refuse to make reviews. And so we have to balance out …these two extremes. (Editor-in-chief, specialty journal).

Lastly, while peer reviewers were expected to fulfil the previous outlined criteria to some degree, interviewees did not consider the completion of a training or a course

on peer reviewing as a prerequisite or necessary qualification to become a peer reviewer. All interviewees stated that they learnt to peer review manuscripts 'by just doing it', without having had previous training, and suggested that this was also the case for the majority of the peer reviewers in biomedical journals. Journal editors explained how one way of honing reviewing skills is through indirect feedback and comparisons with fellow reviewers' reports (ie, operationalised through comparing their own feedback with that of other peer reviewers for the same manuscript) and through the final decision taken by the editor-in-chief on the fate of the manuscript.

> We also tried to train our reviewers in an indirect way that is when a decision was completed and when we send the decision letter to the author we usually carbon copy the decision along with the comments of all the reviewers to all the reviewers so that every reviewer can see and compare their comments, their own comments with the comments of other reviewers and that would be a form of training for them. (Editor-in-chief, specialty journal).

There was a division of opinions on the usefulness of courses that aim to teach peer-reviewing skills. While several editors were receptive to the idea, others felt that they could only be useful to less experienced researchers because they can only teach about the technicalities of the process and cannot replace experience gained over time:

> I learnt on the field. First, as an author and then, you know, when I become more established a scientist, as a reviewer it is a long process, and difficult process… (with) courses, you can learn the technicalities of the process but you know experience is very relevant… courses do not help established scientists, they may help young scientists but the courses won't give them experience in actually in the field. (Editor-in-chief, specialty journal).

### Peer reviewers should be 'dutiful towards scientific community versus volunteers who deserve recognition'

The majority of interviewees repeatedly expressed their gratitude towards peer reviewers, whom they most commonly framed as volunteers who perform peer review out of 'altruistic motives'. Being occasional reviewers themselves, journal editors were well aware of the many competing duties of peer reviewers in the biomedical field—including research, teaching and/or clinical responsibilities—between which reviewing has to be squeezed in. Many interviewees emphasised that reviewing is 'time-consuming' and repeatedly described it as an 'unpaid' and largely 'unrecognised' role:

> Most of the work that is done on journals is uncompensated, and … you are already dealing with people who are very busy people in their professional lives, and so you are really asking them to do things

at nights and weekends for which they get really very little recognition. And very little compensation if any. (Editor-in-Chief, specialty journal).

Given that the majority of journal editors face difficulties finding peer reviewers, several considered peer reviewers to be a 'precious resource' that needs to be treated with 'care'. Interviewees reported doing so through careful screening of submissions to ensure that only sufficiently good-quality manuscripts are forwarded to peer reviewers, not overburdening good peer reviewers with too many invitations, and provision of recognition and rewards. Several recognition and reward schemes were mentioned, which can be broadly divided into two categories : (1) financial rewards (free access to journal/publication discount) and small tokens of appreciation (eg, mugs, books) and (2) rewards aimed at boosting career progress through official professional development (eg, continuing medical education points; official letters for continuing professional development; and through journal rewards aimed at enhancing peer reviewers' visibility, reputation and credibility within the scientific community (eg, being invited to become editors and/or editorial board members, names published on journal website and invitations to social events).

In contrast to the more common perception of reviewers as 'volunteers', a small number of editors commented that peer reviewers should consider the act of peer reviewing to be a 'responsibility', 'duty' and 'obligation to their field' and to the scientific community in general. In their view, the entire process relies on—and only works because of—the principle of reciprocity and researchers perpetuating the development of their own research community. In their view, reciprocity should be a strong motivational drive for peer reviewers:

> Those of us who have a track record in publication get solicited for doing an awful lot of reviewing and you have got to fit that in around your other time and you are doing it because the process is important and you want your next paper to get properly reviewed so you want to peer review the paper that you have been sent. (Interim editor-in-chief, specialty journal).

A few editors were more critical of the rationale for reviewing 'for free', suggesting that the concept of duty in peer reviewing had originally been coined and continued to be fostered by publishers for profit-making purposes and is now dated:

> I mean they… they say this is your duty, you know it is your duty as a scientist to, you know, do these things … and give back, but … really the journals … certainly are profiting now the authors are paying pretty good page charges, the reviewers aren't getting paid, and you know this could be an issue. (Editor-in-chief, specialty journal).

### Peer reviewers should be 'professionals'

There was general agreement on the need for reviewers to be (1) unbiased and ethical professionals, (2) reliable professionals and (3) skilled critics.

Editors outlined three aspects related to their expectation that peer reviewers should be 'unbiased and ethical professionals', consistent with 'scientific ideals'. These were (1) being 'fair' and 'objective' (ie, peer reviewers are expected to evaluate and judge manuscripts in a fair and objective manner); (2) 'maintain confidentiality' (ie, peer reviewers are expected keep manuscript content confidential avoiding disclosure to others); and (3) 'declare/avoid potential or actual conflict of interest'. Editors emphasised the importance of the latter most frequently. Some editors explained that conflict of interest could potentially contribute to increased review quality but stressed that transparency is key. They emphasised their own position as 'decision makers' within the peer-review process to assess and decide whether the reported conflict of interest is prohibiting a fair and objective assessment.

Journal editors also unanimously agreed that peer reviewers should be 'reliable professionals' who should 'respond promptly to peer-reviewer requests'. They should either accept or decline, but not 'ignore the invitation to review', which is the more common frustrating practice reported by interviewees from non-high-ranking journals. The common understanding among all editors was that a good peer-reviewer report takes a substantial amount of time to be written, something that peer reviewers should be aware of prior to accepting. They should be willing to devote sufficient time and attention to the evaluation of manuscripts yet deliver the reviewer report within the agreed timeline out of 'respect' and 'fairness' to authors, to the journal and the publisher.

Lastly, the majority of interviewees considered helping authors to 'improve their manuscript' to be the primary purpose of the peer reviewer, not to suggest a rejection or to 'filter it out'. Therefore, the need for reviewers to be 'skilled critics' was explicitly and implicitly voiced throughout the interviews. As part of the improvement role, it was expected that peer reviewers provide 'constructive criticism embodying specific and addressable comments'. Peer reviewers were also expected to be 'thorough and detailed' and to 'systematically address every aspect of the manuscript'. Another aspect emphasised by interviewees was the need for an 'evidence-based review', where peer reviewers' statements should be 'supported by references' that aid the author and guide the editor.

Journal editors expected peer reviewers to be 'respectful communicators'. They outlined basic principles of courtesy, such as 'respect for the work of the authors'. Peer reviewers were expected to provide comments that 'serve a scientific purpose' while keeping in mind that they are criticising the manuscript, not the authors. Appropriate communication was deemed to be crucial. Based on editors' accounts that peer reviewers should be 'kind' and offer 'positive' comments to nurture and 'encourage'

authors to improve their work, it became evident that peer reviewing should go beyond the mere technical assessment of manuscripts and thus has also a supportive role:

> I often think the peer reviewers are incredibly negative, and they rarely have anything positive to say. And I tend to feel, you know if somebody was reviewing my manuscript I would want them to try to say at least one tiny little positive thing about what I have done. (Editor-in-chief, specialty journal)

### Peer reviewers should be 'advisors to the editor'

Journal editors were explicit in their attribution of a primarily 'advisory role' to peer reviewers. Our interviewees perceived and stressed their own role as the 'ultimate decision makers' who take decisions based on the sum of the factors outlined earlier. They have the authority to 'override peer reviewers recommendations' and 'ignore their opinion', if necessary, thereby directly or indirectly exerting influence on authors to modify their manuscripts:

> …the peer reviewer is really playing an advisory role to the editors…it's only the editors that make a decision on whether to accept or not and how they want the paper to be written. (Editor-in-chief, specialty journal).

Journal editors made it clear that decision making within the editorial process is shaped and influenced by the interplay of a complex web of factors, including (1) the editors' own expert knowledge and ability to assess different aspects of manuscripts, (2) peer-reviewer reports, (3) authors' replies, (4) discussions between editors and editorial board members during manuscript meetings where manuscripts considered for publication are discussed, (5) the number and type of submissions received, (6) the strategic approach of the journal, (7) consideration of readership and (8) subjects related to publishers. Thus, while peer-reviewer reports play a key part, they are not the only element within the equation. While scientific quality and the value of submitted manuscripts were at the foreground, interviewees were largely open about the influence of other non-scientific factors that play into their decision-making process. Nevertheless, the peer-reviewer report was consistently regarded as a key pillar supporting publication decisions, including peer reviewers' advisory role of providing the editor with a 'recommendation on the fate of the manuscript'. With few exceptions, most journal editors reported that their journal submission systems ask peer reviewers to indicate whether the manuscript should be accepted (with major/minor revisions) or rejected:

> …the most important thing for me is actually at the end, the advice to reject the paper or have it revised. (Editor-in-chief, specialty journal).

Most journal editors were open about the substantial influence of peer-reviewer recommendations on their decision making. This was rationalised in a variety of ways, which often coexist. Journal editors partly deferred their decision to peer reviewers when they felt uncertain about their own knowledge and ability to assess the manuscript adequately, referring to the 'trust' they extend towards experts in the field to help in decision making. Ticking the recommendation box was also useful to justify editorial decisions to authors when the peer-reviewer report did not convey a clear direction for the manuscript, and the journal editor wants them to 'come off the fence'. Many editors reported deferring to additional peer reviewers in case of disagreements between the initially selected peer reviewers, described as a common occurrence. Another problematic aspect of the recommendation function was the lack of a common understanding of what the individual recommendation categories actually mean. Since this is a subjective recommendation, there are inherent variations in reviewers' views.

### Peer-reviewer tasks

Journal editors outlined a number of tasks that coalesced around four domains: (1) organisation and approach to reviewing, (2) making general comments, (3) assessing and addressing content for each section of the manuscript, and (4) addressing ethical aspects.

### Organisation and approach to reviewing

At the beginning of the reviewer report, journal editors prefer to see a 'summary of the key points' of the manuscript, which functions as a 'quality check' for editors 'to be confident that they (the peer reviewers) have read it and understood it (the manuscript)'. The majority of journal editors expect reviewers to provide a balanced view by identifying both 'strengths and weaknesses of the manuscript'. Editors also expect peer reviewers to 'identify flaws' and differentiate between 'fatal and addressable flaws' in order to understand and assess whether the manuscripts could be improved. Furthermore, a number of journal editors suggested that it is helpful to differentiate between 'major and minor comments'. It became evident that the approach to peer review is mostly aimed at helping journal editors in their decision-making process.

### Make general comments

Journal editors specified that they expect to see some general and overarching comments that provide an 'overall picture' of the 'importance and significance' of the manuscript, as well as 'relevance to field and (clinical) practice'. Additional comments should focus on the general aspects of 'validity', 'quality', 'technical merit' and 'rigour'. The assessment of 'novelty' and 'originality' was mentioned by a number of editors; however, there was a clear divide between high-ranking journals and other journals, with editors from the latter repeatedly acknowledged that manuscripts with 'novel findings' tend to be preferentially submitted to high-ranking journals.

## Assess and address content for each section of the manuscript

The majority of journal editors expected peer reviewers to thoroughly appraise the content of each manuscript section. The 'soundness of the methodology used' was most frequently mentioned by peer reviewers. Generally, the level of detail expected of peer reviewers seemed to differ according to the resources that journals had, as well as the editors' own abilities. While this was often-times implicit, it was apparent in the example of 'statistics'. For example, while a number of journals reported to employ a 'statistical review by default' other had to rely heavily on peer reviewers for that to supplement their own limitations:

> …bringing expertise such as looking at the statistical analysis which is not my strong point at all. So bringing that sort of expertise to it. (Coeditor-in-chief, specialty journal).

Another aspect that was repeatedly mentioned was the focus on 'spin' in the discussion/conclusion section. Although not explicitly named as spin, editors want peer reviewers to look out for any 'claims that are not supported by the results', 'overenthusiasm' and 'extrapolation'.

## Address ethical aspects

Journal editors reported that their submission systems typically offers two text boxes to peer reviewers: one for comments to the authors and the other one for confidential comments to the editors. The latter should be used by peer reviewers to advise the journal editor on any aspects related to 'ethics' and 'research integrity', such as suspicion of research misconduct and detrimental and questionable research practices. The confidential comments are a means of avoiding any potential conflict arising from such criticism between authors and reviewers.

## DISCUSSION

This study provided an in-depth, behind-the-scenes account of 56 journal editors' experiences with, and expectations towards, peer reviewers. We found that journal editors' understanding of the roles and tasks of peer reviewers are profoundly shaped by each journal's unique context and characteristics, including financial and human resources and journal reputation. Thus, in line with existing literature, we found that editorial decision making and expectations towards peer reviewers are unavoidably shaped by social externalities that at times may have little to do with the scientific content of the manuscript.[6 17] We found that the majority of our interviewees gave considerable importance to the reviewers' recommendation function, despite concerns regarding the lack of a commonly agreed-upon definition of the available options, frequent disagreement among peer reviewers[18] and existing bias.[19] Given these limitations, journal editors should seriously consider removing the reviewers' 'recommendation function', where they are expected to provide the editor with their recommendation

regarding the article's suitability for publication. This is in line with existing research on the relationship between external reviewers' recommendations and the editorial outcome of manuscripts.[20] This would help to realign the role of peer reviewers as 'advisors' rather than convey the idea that they are decision makers. It would also help to delete some of the existing malleable boundaries of authority and responsibility on the review process placing the journal editor in the sole decision-maker position. Considerable efforts should be made to communicate to peer reviewers to place their focus on the evaluation of strengths and weaknesses, major and minor flaws of manuscripts across multiple dimensions, and suggestions for improvement. Furthermore, journal editors should encourage peer reviewers to refer to appropriate reporting guidelines to ensure the completeness of information provided by authors in their studies. One way of achieving this could be through provision of feedback to peer reviewers by journal editors; that is, editors could send follow-up emails to peer reviewers requesting clarification of any missing points. This is time-consuming but might help to improve peer-reviewer reports.

Furthermore, although we found considerable agreement among editors concerning technical tasks of manuscript reviewing, there was an apparent difference in journal editors' expectations of the level of depth and detail they would like to see in a reviewer report. Our study sample showcases the status quo of the journal editors' market, where there are a few full-time journal editors. The remainder work on a part-time basis, usually for a symbolic or stipend-like payment, and combine their editorial responsibilities with research, education and/or clinical duties. Therefore, it might be the case that their own limited time might lead to expectations of greater detail from reviewers. Journal resource availability might also have an impact on their expectations, such as requests for comments related to statistical analysis in the case of journals with fewer resources. Given these existing contextual journal differences and hence peer-review report requirements, better ways of communicating editorial expectations to peer reviewers (who might review for different journals having different expectations) are needed. Currently, these expectations are communicated through publishers and journal-specific guidelines. However, various studies in this area suggest that these are often not readily available, or are generic and non-specific[21] and thus do not properly convey expectations.

Another key finding was interviewed journal editors' apparent lack of appreciation of the importance of formal peer-reviewer training. The majority embraced a somewhat simplistic and 'linear' view that 'good' authors (ie, usually defined as authors with extensive authorship in prestigious journals) make 'good' peer reviewers. However, there is no evidence to support this perspective; evidence linking authorship experience and academic qualifications to high-quality reviews is very limited. The only substantial study in this field was unable to predict reviewer performance from easily identifiable types of

experience or qualifications. The study authors also found that, contrary to the beliefs prevalent among our interviewees, factors such as academic rank and seniority do not predict performance.[22] In fact, studies that have attempted to determine whether some combination of peer-reviewer experience could predict the quality of their subsequent reviews found that the highest-rated reviewers tended to be young and that the quality of peer review did not correlate with academic rank.[23–26] However, most of these studies were relatively limited in size, were a subanalysis of a study of some other intervention and were more than 20 years old; hence, the evidence base for this finding is limited. Thus far, in the absence of additional research demonstrating the contrary, there are no criteria that predict good peer-reviewer performance.

Given this situation, we believe that the skillset required to be a good author is not necessarily the same as that required to be a good peer reviewer. In a recent study (2019) by Superchi *et al* that systematically reviewed tools used by journal editors to assess the quality of peer-review reports, the authors identified nine quality domains pertaining to peer-reviewer skills, of which five (ie, relevance and originality of the study, interpretation of study results, strength and weaknesses, manuscript presentation and organisation) arguably overlap with the skillset of authors. The remainder are directly concerned with skills related to structure and delivery of the peer-review report,[27] which we believe may not automatically follow from being a prolific author. Therefore, we propose that the following four domains can, and in principle should, be taught to prospective reviewers: (1) structure of the reviewer's comments; (2) characteristics of reviewers' comments, including concepts such as clarity, constructiveness, detail/thoroughness, fairness, knowledgeability and tone; (3) timeliness of the review report; and (4) usefulness of the review report to editorial decision making and manuscript improvement. Thus, it appears that helping to improve the manuscript entails providing not only specific and detailed comments about scientific aspects of the manuscript but also comments that empower and motivate authors, a skill that is closely aligned to the supportive function of peer reviewers that also emerged from our study.

Notwithstanding various surveys on the educational needs of young clinicians and researchers across different biomedical fields having revealed a strong interest in attaining better reviewing skills,[28] such training is still not commonly included in biomedical postgraduate education programmes. At the same time, existing educational interventions have shown underwhelming results, and their wider applicability remains questionable due to their relatively poor methodological quality.[29]

Given this lack of evidence, we think it would be helpful to conduct research on the actual content of peer reviewers' reports to help establish educational needs for peer reviewing.[30]

According to the majority of our interviewees, it is becoming increasingly difficult to find experienced authors to review manuscripts. On the other hand, junior researchers are often more willing to accept invitations, including those from lower-ranking journals. This is in line with existing evidence[31] and is likely to be due to differing levels of motivation.[32] Thus, there is an opportunity for acknowledging that the breadth and variety of reviewing roles and tasks may require a more granular approach by editors when assigning peer reviewers to a manuscript. Achieving a balance of senior and junior reviewers would cater to their wide range of reviewing motivations, as well as to their individual expertise. At the same time, the question of how to attract this ideal mix of reviewers remains. The rewards and incentives offered by most journal editors among our sample are likely to be more attractive for junior peer reviewers than senior reviewers. Based on editors' comments on the lack of effectiveness of the provided incentives and the general difficulty to get peer reviewers to accept invitations across the biomedical field[33 34] and offering higher-level rewards is key. For example, the majority of reviewers are affiliated to academic institutions, which are therefore critical stakeholders in the peer-review process. If peer reviewing is incentivised and rewarded as part of one's academic career advancement, it is likely to be as important—if not more important—than whatever journals can offer. For example, the University of Glasgow[35] has started rewarding peer-reviewer and editorial responsibilities as a core requirement for academic promotion and achieving tenure. However, this is the only example we were able to identify. The peer-review process is part of the social infrastructure of research[36]; therefore, it is the responsibility of all actors to contribute to better research. Academic institutions and other stakeholders such a funders can play a key role to implement alternative measures of research quality[37] and a stronger focus on research quality.

## LIMITATIONS

Our recruitment approach gave rise to a key limitation of this study. Based on our collective experience as researchers and a former staff member of a biomedical journal (DH), who struggled with response rates involving studies with editorial staff, we anticipated that it would be challenging to recruit journal editors to participate in our research. The majority of journal editors of biomedical journals are part-timers who concurrently work as practitioners, researchers and educators and may have other additional roles. In the light of this situation, our employment of purposive maximum variation sampling resulted in predominant contact with editors-in-chief. While one of the strengths of this study is that research participants were diverse in terms of demographic characteristics and characteristics related to their journal (table 1), two-thirds of participants had this role within their respective journal. Although the lead researcher asked potential interviewees either to participate themselves or to recommend suitable journal colleagues who could be contacted in their stead, it is likely that this approach led to the

relative homogeneity of our study sample. This may limit the generalisability of the results due to the limited representation of other editorial staff members involved in the peer-review process. Our insights from the interviews and wider author and team experiences suggest that editors-in-chief might primarily be responsible for higher-level tasks around the journal, and possibly be less involved in the direct communication process with authors and peer reviewers. Therefore, there is a need to explore whether the involvement of editorial staff in other positions would have produced convergent or divergent findings.

## CONCLUSION

This study provides context for, and details about, the roles and tasks of peer reviewers in biomedical journals and helps to explain attitudes and opinions expressed in existing surveys of editors, reviewers and authors on the peer-review process. Our research provides a greater understanding of the current status quo of the review process and why particular issues arise around roles and tasks of peer reviewers, and offers insight into how these issues can be addressed.

Further research is needed on actual performance of peer reviewers looking into the content of peer-reviewer reports on a large scale to inform meaningful training interventions and to improve existing journal policies and guidelines.

**Acknowledgements** The authors thank Dr Sara Schroter (British Medical Journal) and Dr Elizabeth Moylan (BioMed Central) for providing guidance and help on the recruitment strategy of interviewees; the publishers and all the study participants; and also Alice Biggane (PhD candidate on the Methods in Research on Research project) for her help in reviewing this manuscript prior to submission.

**Contributors** All authors have made substantive intellectual contributions to the development of this manuscript. KG and DH jointly contributed to the study conception and design, while KG led the data collection, analysis and writing of the manuscript; DH led the supervision of all these steps. IB and DM have contributed to the writing of the manuscript and approved the final manuscript.

**Funding** This project was supported by the European Union's Horizon 2020 research and innovation programme under the Marie Sklodowska-Curie grant agreement number 676207.

**Competing interests** KG and DM had an advisory role with Publons Academy. At the time of data collection for this study, KG conducted a secondment at the BMJ. The remaining authors (IB and DH) declare no competing interests.

**Patient consent for publication** Not required.

**Ethics approval** This project has been evaluated and approved by the University of Split, Medical School Ethics Committee. Ethical approval (reference number 2181-198-03-04-17-0029) was granted in May 2017. Prospective interviewees were provided with a study consent form and a study information sheet. Interviewees were asked to sign a written consent form prior to being interviewed. Copies of the invitation letter, information sheet and consent form are available from the leading author (KG).

**Provenance and peer review** Not commissioned; externally peer reviewed.

**Data availability statement** Data are available upon reasonable request. The data generated and/or analysed in the study are not publicly available due to participant anonymity, but may be available from the corresponding author on reasonable request, which include a study protocol, ethical approval and data use agreement.

**ORCID iDs**
Ketevan Glonti http://orcid.org/0000-0001-9991-7991
Isabelle Boutron https://orcid.org/0000-0002-5263-6241
David Moher http://orcid.org/0000-0003-2434-4206
Darko Hren http://orcid.org/0000-0001-6465-6568

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
