## [Reviewer comments · BMJ Open]

ARTICLE DETAILS

TITLE (PROVISIONAL)	Editors' perspectives on the roles and tasks of peer reviewers in biomedical journals: A qualitative study
AUTHORS	Glonti, Ketevan; Boutron, Isabelle; Moher, David; Hren, Darko

VERSION 1 – REVIEW

REVIEWER	Armen Yuri Gasparyan Dudley Group NHS Foundation Trust, Dudley, UK
REVIEW RETURNED	16-Aug-2019

GENERAL COMMENTS	The authors present an interesting study based on interviewing editors employed by large publishers and those attending a global peer review congress. The Limitations are transparently presented. I have some comments. 1. The Introduction could be expanded to justify the novelty of the current study. The following relevant articles could be consulted: https://www.ncbi.nlm.nih.gov/pubmed/22911533 https://www.ncbi.nlm.nih.gov/pubmed/310158832. The Aim needs to be more specific about who were interviewed - editors of general medical, specialized, or both?3. How many editors were contacted totally, and how many did not respond?4. Sample characteristics. It would be appropriate to report the indexing status of the journals in Scopus, Web of Science and MEDLINE. Listing impact factor as a journal characteristic is not recommended. Additionally, listing the editors' membership in the Committee on Publication Ethics is advisable. The COPE provides guidance on peer review which is globally applicable.5. Limitations. Discuss why the role of peer reviewers in enforcing the authors' adherence to the EQUATOR Networks standards.
---

REVIEWER	Stuart T Haines, Pharm.D. University of Mississippi School of Pharmacy Jackson, Mississippi USA
REVIEW RETURNED	22-Aug-2019

GENERAL COMMENTS	A well-written and methodologically sound research paper. The investigators conducted semi-structured interviews with 56 journal editors to determine the perceived roles and tasks that peer reviewers are expected to fulfill. This information has useful applications toward improving the quality of peer review and the editorial "ecosystem" for scientific / biomedical journals. A few minor changes would enhance the paper's clarity and transparency.
--

	ABSTRACT and INTRODUCTION. No comments or suggestions for improvement. METHODS The authors are to be commended on their strong methodologic and analytical approach. However, there are few items on the consolidated criteria for reporting qualitative studies (COREQ) that were not reported. In the absence of a clear justification why these items can not be reported (or why it was not performed), I believe these can be easily included in the revised manuscript. Specifically:  1) Describe how the relationship with the participants was established. Currently, the recruitment strategy likely provides sufficient information but may some clarification. 2) Participant knowledge of the interviewer. If the interview did not share any information about the investigators' goals and reasons for conducting the study, this should be clearly stated in the methods. Otherwise, a brief statement about what the participants were told in advance of their participation is important because this information would obviously frame the context of the interview and may have (inadvertently) shaped the participants' answers. 3) Repeat interviews. Based on a description of the research methods, I don't believe repeat interviews were conducted and if so, please make an explicit statement to that effect. If some participants were interviewed more than once, it's important to report how many and why. 4) Description of the coding tree. If the investigators did not develop a coding tree, this should be explicitly mentioned in the paper. 5) Participant checking. If the participants were not provided an opportunity to provide feedback on the findings, this should be explicitly stated. However, I believe this is a best-practice that could enhance the legitimacy and rigor of the final report. Therefore, I encourage the investigators to provide an opportunity for their participants to provide feedback regarding this report (before final publication) to ensure that important concepts have not been misinterpreted. RESULTS Roles of peer reviewers - Peer reviewers should be "Proficient Experts." Page 5, Line 59. The word "unapologetic" seems like an editorial comment. Unless the panelist literally did not apologize for prioritizing what journals they elect to accept reviews for based on the journals reputation / impact, I think this kind of (subjective) discriptor doesn't belong in the results section. Roles of peer reviewers - Peer reviewers should be "Professionals." The "professional" role description left me wondering if this theme included some "tasks", rather than being a role. The findings clearly indicate that editors expect peer reviewers to respond to requests in a timely manner, complete their reviews in the agreed time frame, hold communications confidential, and to communicate in a respectful manner. These findings don't seem to group well together and I'm not sure "Professional" is the best descriptor. The first three in this list seem more "task" oriented and exemplify "professionalism" (the outward behaviors of professionals). The final role described in this section relates to "improving the manuscript." This seems to be an important role similar to "advisor to the editor" - a peer reviewer
--	---

	has role obligations to the author(s) to be a "trusted colleague" or "skilled critic" who provides constructive feedback. This is a very different role / task than the other 3 concepts covered in this theme. DISCUSSION Page 11, Lines 37-38. "Given these limitations, editors should seriously consider doing away with this requirement." I had to read this line several times to figure out what "requirement" was being alluded to here. My assumption is the "requirement" to provide the editor with a publication decision. Clarifying the statement would be helpful. Appendices / Tables / Figures I commend the authors for including the Topic Guide for Semi-structured Interviews with their paper as well as the Actions Undertaken to Establish Trustworthiness of Analysis with the paper. These increase the transparency and replicability of the study.
--	--

REVIEWER	Amy Price University of Oxford UK Stanford University USA The BMJ and The BMJ Group UK I am a BMJ editor
REVIEW RETURNED	01-Sep-2019

GENERAL COMMENTS	This work notes that "Peer reviewers of biomedical journals are expected to perform a large number of roles and tasks, some of which are seemingly contradictory or demonstrate incongruities between the respective positions of peer reviewers and journal editors" and this is an interesting area of focus. Not having clearly defined roles can be frustrating and confusing. The aim of this study was to explore the perspectives of journal editors regarding the roles and tasks of peer reviewers. There was considerable focus on a prior scoping review which is distracting in some places as the claims are rather general. The study design was qualitative with participants being 56 journal editors from biomedical journals, They were interviewed and the themes iteratively developed until saturation was reached. Editors broadly agreed on the technical tasks of peer reviewers but show different expectations in the level of depth. Editors perspectives favoured experience over training. There were missed opportunities for journal editors to engage with peer reviewers to clarify the expected roles and tasks. The authors conclude social dimensions of biomedical manuscript review should be made more explicit and targeted feedback shared to improve future review. The authors worked hard on this manuscript as it is difficult to bring so many divergent voices into a whole and write it up. They make interesting and insightful points. Below are areas where the work in my view might be strengthened. Introduction
---

	Might the authors provide a more direct and succinct connection between the introduction and the discussion? “The aim of this study was to explore the perspectives of journal editors regarding the roles and tasks of peer reviewers” Perhaps the discussion could focus here as other areas tend to dilute the aim. Methods Patient and Public Involvement It was not appropriate or possible to We did not involve patients or the public in the design, or conduct, or reporting, or dissemination of our research. PPI was appropriate for several reasons, Public and patient review is on the rise and increasingly common in grant writing, proposal review, manuscript review and even IRB review. The public are readers or users of the manuscripts that are reviewed, they might have helped to consider your questions for the editors, to help you in disseminating the work, to assess your work for readability and to suggest areas the authors may have missed. Please reconsider this wording. It may be helpful to build a table or figure with the domains for each theme and how many contributed to this. This can be easily done in NVIVO and is a clear transparent way to share your work. Results Peer reviewers should be ‘Professionals’ Of the nine domains identified regarding the role of peer reviewers there was general agreement on the need for reviewers to be: (1) Unbiased and ethical professionals; (2) Reliable professionals; (3) Skilled critics What are the nine, best to list them all and then say there were these 3 with the general agreement or skip mentioning the nine and focus on the three? Peer reviewers should be ‘Advisors to the editor’ This was interesting and it would be even more powerful to review the reasons for this. For example, reviewers should never be held accountable, ridiculed or bullied for their review by the authors and if these lines were clear with reviewers as valued and respected advisors and there were consequences for bad behaviours by authors, these behaviours would be less common and it would give the reviewers more psychological safety. Editors of lower-ranking journals felt resentful at this “arrogant” and “dismissive” attitude, and yet were cognisant of the lower “quality” of the research that they receive and end up publishing in comparison with high-ranking journals until their journal gains “visibility” through external recognition in the form of an impact factor and becomes “attractive” to authors that offer better quality research: This is not congruent with the perspectives and tasks of peer reviewers and this inter editor complaining is a distraction. In contrast, since the motivation for reviewers to deliver a high-quality report in high ranking journals was to maintain their status and reputation within the ‘elite’ scientific community, their reviewing standards are likely to be different than those for lower-ranked journals: This is an assumption, as an editor with a high impact journal who has seen reviews by the same reviewers to low and high impact journals I would say it is inaccurate. Reviewers review the way they do because that is what they have learned to do through feedback absent or present and that is their personality and skill set. It is fine to quote an editor’s perspective as this is what the research
--	--

	is about. When an author assumes this posture by accepting this as “truth” it is disturbing, I would suggest deleting. Discussion Some academic institutions, for example, the University of Glasgow (34) have started rewarding peer reviewer and editor responsibilities as a core requirement for academic promotion and achieving tenure. Are there others? https://journals.plos.org/plosbiology/article?id=10.1371/journal.pbio.2004089 https://www.ncbi.nlm.nih.gov/pmc/articles/PMC6325612/ Given these limitations, editors should seriously consider doing away with this requirement altogether. Instead, considerable efforts should be made to communicate to peer reviewers to place their focus on the evaluation of strengths and weaknesses, major and minor flaws of manuscripts across multiple dimensions and suggestions for improvement. How will they prepare reviewers for this task, most instructions to authors ask that reviewers do this already? Acknowledgements Was assistance given during the secondment to The BMJ? Were their individuals who assisted in areas that helped the development of this work and if so it might be appropriate to acknowledge them. Would it be appropriate to acknowledge and thank the publishers who shared editors for this project?
--	--

VERSION 1 – AUTHOR RESPONSE

Reviewer #1:

The authors present an interesting study based on interviewing editors employed by large publishers and those attending a global peer review congress. The Limitations are transparently presented.

- We thank the reviewer for the positive feedback.

1. The Introduction could be expanded to justify the novelty of the current study. The following relevant articles could be consulted: <https://www.ncbi.nlm.nih.gov/pubmed/22911533>, <https://www.ncbi.nlm.nih.gov/pubmed/31015883>

- Thank you for this relevant suggestion and the articles provided. We have now expanded the introduction (page 2) accordingly.

“A recent scoping review (2019), showed that there is a large number of roles and tasks that peer reviewers of biomedical journals are expected to carry out some of which seemed to contradict each other, and apparent or displayed incongruities between the position of the peer reviewer and the position of the journal editor (3). These findings were reflected in a study that aimed to identify the tasks that journal editors expect from peer reviewers who evaluate a manuscript reporting a randomised controlled trial, where a substantial disconnect between the expectations of journal editors and peer reviewers was found (4). A mutual understanding of expectations and responsibilities is one of the key factors that determine the quality of reviewer reports and satisfaction of the actors with the review process. However, biomedical journals differ in their guidance provided to peer reviewers, in their publishing capacity and resources available as well as the reviewer pool (5). Therefore, it is likely that editors might have diverging opinions about the roles and tasks peer reviewers are supposed to perform, something that has not been previously explored in depth. Given that peer review is a complex social process that goes beyond the quality control of manuscripts (6), qualitative methods may lead to a deeper examination of the complexities of these

processes compared to quantitative approaches and provide important context to improve the understanding of different editorial realities and practices.

Our aim was to examine the experience of general and specialty biomedical journal editors and to characterise their perspectives, expectations and understanding of the roles and tasks of peer reviewers.”

2. The Aim needs to be more specific about who were interviewed - editors of general medical, specialized, or both?

- We thank the reviewer for raising this point. We agree that our aim needs to be more specific. We have now updated it accordingly (page 2).

“Our aim was to examine the experience of general and specialty biomedical journal editors and to characterise their perspectives, expectations and understanding of the roles and tasks of peer reviewers.”

We have also refined the first sentence in the ‘Methods’ section (under the subheading ‘Study design’, page 2) to reflect this suggestion.

“We conducted semi-structured interviews with biomedical journal editors from general and specialty journals.”

3. How many editors were contacted totally, and how many did not respond?

Thank you for this comment. We have now added this information into the “Sampling and recruitment section” (page 3) and re-written this paragraph accordingly.

“A total of 543 prospective interviewees were approached via email and 69 editors responded positively to the request. In addition, interviewees were asked to recommend other editors who would potentially be interested in contributing to this study.

Since sample size is irreversibly linked to saturation, which in turn can only be operationalized during data collection (12), our approach to data collection and analysis was iterative. Thus, recruitment continued until saturation – conceptualized as the point at which no new codes and themes were identified from the data – was achieved. After 56 interviews saturation was obtained and no further editors were contacted and interviewed.”

4. Sample characteristics. It would be appropriate to report the indexing status of the journals in Scopus, Web of Science and MEDLINE. Listing impact factor as a journal characteristic is not recommended. Additionally, listing the editors' membership in the Committee on Publication Ethics is advisable. The COPE provides guidance on peer review, which is globally applicable.

- We thank the reviewer for raising these relevant points. We agree that it is appropriate to report the indexing status of the journals as well as the editors' membership on the Committee on Publication Ethics, and have updated “Table 1. Sample characteristics” (p. 4) accordingly. We have also deleted the information on the impact factor.

5. Limitations. Discuss why the role of peer reviewers in enforcing the authors' adherence to the EQUATOR Networks standards.

- Thank you for this suggestion. We have now added this aspect into our “discussion” section (page 11-12).

“Furthermore, editors should encourage peer reviewers to refer to appropriate reporting guidelines to ensure the completeness of information provided by authors in their studies.”

Reviewer #2:

A well-written and methodologically sound research paper. The investigators conducted semi-structured interviews with 56 journal editors to determine the perceived roles and tasks that peer reviewers are expected to fulfil. This information has useful applications toward improving the quality of peer review and the editorial "ecosystem" for scientific / biomedical journals. A few minor changes would enhance the paper's clarity and transparency.

- Thank you for the positive feedback. We have undertaken a number of changes to enhance the paper's transparency in accordance to your comments.

ABSTRACT and INTRODUCTION. No comments or suggestions for improvement.

- We have undertaken minor changes in the abstract and introduction based on the suggestions of the other two peer reviewers.

METHODS: The authors are to be commended on their strong methodologic and analytical approach. However, there are few items on the consolidated criteria for reporting qualitative studies (COREQ) that were not reported. In the absence of a clear justification why these items cannot be reported (or why it was not performed), I believe these can be easily included in the revised manuscript.

1) Describe how the relationship with the participants was established. Currently, the recruitment strategy likely provides sufficient information but may some clarification.

Thank you for this comment. We have added that the participants were approached via email and through snowballing in the "Sampling and recruitment" section (page 3) and added further information in response to your second comment (see below). We have also updated this item in the checklist of the reporting guidelines.

"A total of 543 prospective interviewees were approached via email and 69 editors responded positively to the request. In addition, interviewees were asked to recommend other editors who would potentially be interested in contributing to this study."

2) Participant knowledge of the interviewer. If the interview did not share any information about the investigators' goals and reasons for conducting the study, this should be clearly stated in the methods. Otherwise, a brief statement about what the participants were told in advance of their participation is important because this information would obviously frame the context of the interview and may have (inadvertently) shaped the participants' answers.

- Thank you for highlighting this. We have now added the following paragraph to the "Data collection section" (page 3).

"Prospective interviewees were provided with a study consent form and a study information sheet that consisted of information about the researchers, and study information (aim, interview procedures, ethics, confidentiality, funding and contact details). Interviewees were asked to sign a written consent form prior to being interviewed. Before starting the interview, study objectives were reiterated and additional information provided where necessary."

3) Repeat interviews. Based on a description of the research methods, I do not believe repeat interviews were conducted and if so, please make an explicit statement to that effect. If some participants were interviewed more than once, it's important to report how many and why.

- Thank you for pointing out this lack of clarity. We have now made an explicit note of this in the reporting guideline, specifying that no repeat interviews were carried out.

4) Description of the coding tree. If the investigators did not develop a coding tree, this should be explicitly mentioned in the paper.

- Thank you for this comment. We developed an elaborate codebook with hierarchical structure which is an extended version of a coding tree. The development of our codebook is described in the "Analysis" paragraph (page 3) and in the "Trustworthiness of analysis" (Phase 2: Generating initial codes. We have now made a note of this in the checklist of the reporting guidelines.

5) Participant checking. If the participants were not provided an opportunity to provide feedback on the findings, this should be explicitly stated. However, I believe this is a best-practice that could enhance the legitimacy and rigor of the final report. Therefore, I encourage the investigators to provide an opportunity for their participants to provide feedback regarding this report (before final publication) to ensure that important concepts have not been misinterpreted.

- This is an interesting and contentious point. Participant checking was not performed, nor was it considered to be a key aspect of ensuring trustworthiness of our methods and results. Instead we have used other means to ensure that important concepts have not been misinterpreted, as outlined in detail in the "Trustworthiness of analysis" attachment, for example by multiple returns to raw data to check for referential adequacy by the research team. While the usefulness of participant checking is often up for discussion and has both supporters and detractors, our reading on the matter indicates that it is not considered to be a methodological 'best-practice' in qualitative research. We believe that this stance is amply supported by available literature around qualitative methodology (for example: Peditto K. Reporting Qualitative Research: Standards, Challenges, and Implications for Health Design. HERD: Health Environments Research & Design Journal. 2018 Apr;11(2):16-9). While our results, discussion and conclusion as authors are informed by the sum of all interviews and an overview of the entire dataset, the participants do not have access to this by looking at the end product and cannot therefore make an informed judgement regarding the accuracy of our interpretation.

However, based on your suggestion we have now explicitly noted that participant checking was not performed in the reporting guidelines checklist.

RESULTS. Roles of peer reviewers - Peer reviewers should be "Proficient Experts." Page 5, Line 59. The word "unapologetic" seems like an editorial comment. Unless the panellist literally did not apologize for prioritizing what journals they elect to accept reviews for based on the journals reputation / impact, I think this kind of (subjective) descriptor doesn't belong in the results section.

- Thank you for pointing out this. We agree that this word might be inappropriate here. We have now deleted it.

"Since they typically receive a high volume of reviewer requests, journal editors suspect that they prioritize their reviewing time in favour of highly ranked journals, a behaviour that multiple journal editors reported practicing themselves when asked to perform a peer review."

Roles of peer reviewers - Peer reviewers should be "Professionals." The "professional" role description left me wondering if this theme included some "tasks", rather than being a role. The findings clearly indicate that editors expect peer reviewers to respond to requests in a timely manner, complete their reviews in the agreed time frame, hold communications confidential, and to communicate in a respectful manner. These findings don't seem to group well together and I'm not sure "Professional" is the best descriptor. The first three in this list seem more "task" oriented and exemplify "professionalism" (the outward behaviours of professionals). The final role described in this section relates to "improving the manuscript." This seems to be an important role similar to "advisor to the editor" - a peer reviewer has role obligations to the author(s) to be a "trusted colleague" or "skilled critic" who provides constructive feedback. This is a very different role / task than the other 3 concepts covered in this theme.

- We thank the reviewer for this thoughtful comment. We understand that, given the lack of clarity around this topic in the current literature, assigning duties of a peer reviewer to the concept of 'roles'

or 'tasks' may seem subjective. However, as part of our previously conducted scoping review on the roles and tasks of peer reviewers in biomedical journals (i.e. Glonti K, Cauchi D, Cobo E, Boutron I, Moher D, Hren D. A scoping review on the roles and tasks of peer reviewers in the manuscript review process in biomedical journals. BMC medicine. 2019 Dec;17(1):118.) we embarked on multiple discussions regarding the difference between roles and tasks. Ultimately, we decided to define the term 'roles' as referring to those duties that reflect the overarching nature of peer reviewers' function (Table 1 and 2 of the scoping review) In contrast, we decided to refer to tasks in terms of specific actions that the peer reviewer performs on the manuscript (e.g. commenting on individual sections;). For example, we agree that 'being timely' exemplifies professionalism, as do the other aspects that we defined as roles in Table 1 of the scoping review. Being "Unbiased and ethical professionals", "Reliable professionals", "Skilled critics" (which includes the sub-theme of "improving a manuscript" as you suggest in your comment) and "Respectful communicators" were considered to be among the roles of peer reviewers, and we incorporated this definition and classification into our codebook. DISCUSSION Page 11, Lines 37-38. "Given these limitations, editors should seriously consider doing away with this requirement." I had to read this line several times to figure out what "requirement" was being alluded to here. My assumption is the "requirement" to provide the editor with a publication decision. Clarifying the statement would be helpful.

- Thank you for pointing out this. We agree that it is confusing and have now clarified this statement (page 11).

"Given these limitations, editors should seriously consider removing the reviewers' 'recommendations function', where they are expected to provide the editor with their recommendation regarding the article's suitability for publication. This is in line with existing research on relationship between external reviewers' recommendations and the editorial outcome of manuscripts (19)."

APPENDICES/TABLES/FIGURES - I commend the authors for including the Topic Guide for Semi-structured Interviews with their paper as well as the Actions Undertaken to Establish Trustworthiness of Analysis with the paper. These increase the transparency and replicability of the study.

- Thank you for the encouraging comments.

Reviewer #3:

This work notes that "Peer reviewers of biomedical journals are expected to perform a large number of roles and tasks, some of which are seemingly contradictory or demonstrate incongruities between the respective positions of peer reviewers and journal editors" and this is an interesting area of focus. Not having clearly defined roles can be frustrating and confusing. The aim of this study was to explore the perspectives of journal editors regarding the roles and tasks of peer reviewers. There was considerable focus on a prior scoping review, which is distracting in some places as the claims are rather general. The study design was qualitative with participants being 56 journal editors from biomedical journals. They were interviewed and the themes iteratively developed until saturation was reached. Editors broadly agreed on the technical tasks of peer reviewers but show different expectations in the level of depth. Editors' perspectives favoured experience over training. There were missed opportunities for journal editors to engage with peer reviewers to clarify the expected roles and tasks. The authors conclude social dimensions of biomedical manuscript review should be made more explicit and targeted feedback shared to improve future review. The authors worked hard on this manuscript, as it is difficult to bring so many divergent voices into a whole and write it up. They make interesting and insightful points. Below are areas where the work in my view might be strengthened.

- Thank you for the encouraging feedback. We agree that some of the references to our previous scoping review were not well integrated. We have now removed them from the abstract and modified our references to the scoping review in the introduction and discussion.

Introduction. Might the authors provide a more direct and succinct connection between the introduction and the discussion? "The aim of this study was to explore the perspectives of journal

editors regarding the roles and tasks of peer reviewers” Perhaps the discussion could focus here, as other areas tend to dilute the aim.

- Thank you for this suggestion. We have now expanded our introduction (page 2) to make the link between the introduction and discussion more explicit.

“A recent scoping review (2019), showed that there is a large number of roles and tasks that peer reviewers of biomedical journals are expected to carry out some of which seemed to contradict each other, and apparent or displayed incongruities between the position of the peer reviewer and the position of the journal editor (3). These findings were reflected in a study that aimed to identify the tasks that journal editors expect from peer reviewers who evaluate a manuscript reporting a randomised controlled trial, where a substantial disconnect between the expectations of journal editors and peer reviewers was found (4). A mutual understanding of expectations and responsibilities is one of the key factors that determine the quality of reviewer reports and satisfaction of the actors with the review process. However, biomedical journals differ in their guidance provided to peer reviewers, in their publishing capacity and resources available as well as the reviewer pool (5). Therefore, it is likely that editors might have diverging opinions about the roles and tasks peer reviewers are supposed to perform, something that has not been previously explored in depth. Given that peer review is a complex social process that goes beyond the quality control of manuscripts (6), qualitative methods may lead to a deeper examination of the complexities of these processes compared to quantitative approaches and provide important context to improve the understanding of different editorial realities and practices. Our aim was to examine the experience of general and specialty biomedical journal editors and to characterise their perspectives, expectations and understanding of the roles and tasks of peer reviewers.”

Methods, Patient and Public Involvement “It was not appropriate or possible to involve patients or the public in the design, or conduct, or reporting, or dissemination of our research.”

PPI was appropriate for several reasons. Public and patient review is on the rise and increasingly common in grant writing, proposal review, manuscript review and even IRB review. The public are readers or users of the manuscripts that are reviewed; they might have helped to consider your questions for the editors, to help you in disseminating the work, to assess your work for readability and to suggest areas the authors may have missed. Please reconsider this wording.

- Thank you for pointing this out. We agree with your comment, and have now modified our wording to the following:

“Patients and the public were not involved in the design, or conduct, or reporting, or dissemination of our research.”

It may be helpful to build a table or figure with the domains for each theme and how many contributed to this. This can be easily done in NVIVO and is a clear transparent way to share your work.

- We thank the reviewer for this suggestion. While quantification of themes is not a core aspect of the thematic analysis paradigm described by Braun and Clark that we employed in our study, we followed your advice and included a figure with the domains for each theme. We have made the following note of this on page 5 and included the figure as an additional file:

“Figure 1 displays the domains within our two themes: Roles of peer reviewers and Tasks of peer reviewers (Additional file 3).”

Results. Peer reviewers should be ‘Professionals’. Of the nine domains identified regarding the role of peer reviewers there was general agreement on the need for reviewers to be: (1) Unbiased and ethical professionals; (2) Reliable professionals; (3) Skilled critics. What are the nine? Best to list them all and then say there were these 3 with the general agreement or skip mentioning the nine and focus on the three?

- Thank you for highlighting this. We agree that this is unclear. We have now rephrased this in paragraph (page 7) accordingly:

“There was general agreement on the need for reviewers to be: (1) Unbiased and ethical professionals; (2) Reliable professionals; (3) Skilled critics.”

Peer reviewers should be “Advisors to the editor” - This was interesting and it would be even more powerful to review the reasons for this. For example, reviewers should never be held accountable, ridiculed or bullied for their review by the authors and if these lines were clear with reviewers as valued and respected advisors and there were consequences for bad behaviours by authors, these behaviours would be less common and it would give the reviewers more psychological safety.

- We thank the reviewer for raising this interesting point. While we agree that the example provided would be very interesting to explore in further depth, it is not a finding that emerged from our interviews, hence we do not feel that we are able to comment about it. We believe that delving into this area warrants a separate follow-up study.

Editors of lower-ranking journals felt resentful at this “arrogant” and “dismissive” attitude, and yet were cognisant of the lower “quality” of the research that they receive and end up publishing in comparison with high-ranking journals until their journal gains “visibility” through external recognition in the form of an impact factor and becomes “attractive” to authors that offer better quality research: This is not congruent with the perspectives and tasks of peer reviewers and this inter editor complaining is a distraction.

- Thank you for highlighting this. We agree that this paragraph is not congruent with the aim of the study and have now deleted it.

“In contrast, since the motivation for reviewers to deliver a high-quality report in high ranking journals was to maintain their status and reputation within the 'elite' scientific community, their reviewing standards are likely to be different than those for lower-ranked journals”. This is an assumption, as an editor with a high impact journal who has seen reviews by the same reviewers to low and high impact journals I would say it is inaccurate. Reviewers review the way they do because that is what they have learned to do through feedback absent or present and that is their personality and skill set. It is fine to quote an editor’s perspective as this is what the research is about. When an author assumes this posture by accepting this as “truth” it is disturbing, I would suggest deleting.

- We thank the reviewer for this suggestion. We agree with the spirit of this comment and have now deleted this paragraph.

Discussion. “Some academic institutions, for example, the University of Glasgow (34) have started rewarding peer reviewer and editor responsibilities as a core requirement for academic promotion and achieving tenure”. Are there others?

<https://journals.plos.org/plosbiology/article?id=10.1371/journal.pbio.2004089>,

<https://www.ncbi.nlm.nih.gov/pmc/articles/PMC6325612/>

- Thank you for raising this question. Thus far, we are only aware of this one example. We have now made a note of this in the manuscript (page 13). Although interesting, we didn’t include the study links provided because they discuss existing practices and offer suggestions for assessing scientists and associated research and policy implications, rather than providing actual examples that we could reference.

“For example, the University of Glasgow (34) has started rewarding peer reviewer and editorial responsibilities as a core requirement for academic promotion and achieving tenure. However, this is the only example we were able to identify.”

“Given these limitations, editors should seriously consider doing away with this requirement altogether. Instead, considerable efforts should be made to communicate to peer reviewers to place

their focus on the evaluation of strengths and weaknesses, major and minor flaws of manuscripts across multiple dimensions and suggestions for improvement.“ How will they prepare reviewers for this task, most instructions to authors ask that reviewers do this already?

- Thank you for highlighting the need to further clarify this particular argument. We agree with your suggestion and have now rephrased this paragraph and added further clarification (page 11):

“Given these limitations, editors should seriously consider removing the reviewers' ‘recommendations function’, where they are expected to provide the editor with their recommendation regarding the article’s suitability for publication. This is in line with existing research on relationship between external reviewers' recommendations and the editorial outcome of manuscripts (19). This would help to realign the role of peer reviewers as ‘advisors’ rather than convey the idea that they are decision makers. It would also help to delete some of the existing malleable boundaries of authority and responsibility on the review process placing the editor in the sole decision maker position. Considerable efforts should be made to communicate to peer reviewers to place their focus on the evaluation of strengths and weaknesses, major and minor flaws of manuscripts across multiple dimensions and suggestions for improvement. One way of achieving this could be through] provision of feedback to peer reviewers by editors i.e. editors could send follow-up emails to peer reviewers requesting clarification of any missing points. This is time consuming, but might help to improve peer reviewer reports.”

Acknowledgements. Was assistance given during the secondment to The BMJ? Were their individuals who assisted in areas that helped the development of this work? If so, it might be appropriate to acknowledge them. Would it be appropriate to acknowledge and thank the publishers who shared editors for this project?

- We thank the reviewer for picking up this important point. We provided the suggested acknowledgments in the previously published study protocol but missed the opportunity to do so in this manuscript. We have now added the appropriate acknowledgments to this section.

VERSION 2 – REVIEW

REVIEWER	Armen Yuri Gasparyan Departments of Rheumatology and Research and Development, Dudley Group NHS Foundation Trust (Teaching Trust of the University of Birmingham, UK), Russells Hall Hospital, Dudley, West Midlands, UK
REVIEW RETURNED	01-Oct-2019
GENERAL COMMENTS	I am satisfied with all the amendments. Congratulations.
REVIEWER	Stuart T. Haines University of Mississippi, Department of Pharmacy Practice, USA
REVIEW RETURNED	30-Sep-2019
GENERAL COMMENTS	Thank you for the opportunity to re-review this paper. The authors have made substantive changes to the introduction, methods, and discussion sections of the paper. These revisions, I believe, have adequately addressed the reviewers' comments and concerns.

REVIEWER	Amy Price The BMJ Editorial, University of Oxford, and Stanford University
REVIEW RETURNED	30-Sep-2019

GENERAL COMMENTS	Thank you for your revisions and your courteous response. I wish you the very best with the publication
---